# The Effect of the Manner in Which Montane and Submontane Areas Are Utilized on the Quality of Leachate Water

Piotr Kacorzyk and Jacek Strojny *

Faculty of Agriculture and Economics, University of Agriculture in Krakow, Al. Mickiewicza 21, 31-120 Kraków, Poland; piotr.kacorzyk@urk.edu.pl
* Correspondence: rrstrojn@cyf-kr.edu.pl

**Abstract:** This study aimed at assessing the effect of how submontane soils are managed on the quantity and quality of leachate water, as well as on the load of nutrients leached with it. The quality of leachate water moving through the soil profile at the depth of 0–30 cm was investigated. The quality of leachate water from six research variants was analyzed in three periods: intensive growing, inhibited growing, and the non-growing season. It was established that the type of flora had a significant effect on the amount and chemical composition of water flowing through the soil profile. The highest loads of minerals were leached with leachate waters from arable land. Contrary to the common opinion, unused meadow had the best quality of leachate waters. On account of the quality of leachate waters in submontane and montane areas, it is recommended to reduce plow tillage in these areas. It is also recommended to use these areas as meadows and pastures, with moderate fertilization and rational use, i.e., two mowings or three grazings during the growing season. The study emphasizes how important the management of the use of submontane and montane areas is for the quality and quantity of leachate waters.

**Keywords:** leachate waters; water-bearing areas; crop management; floristic composition



## 1. Theoretical Framework

Montane areas cover approximately 9% of Poland's surface area. The Carpathians (where this study was carried out) constitute two-thirds of montane areas in Poland. These areas perform many important environmental functions, including the hydrological function. Poland is a poor country in terms of water resources. Hydrological importance of montane areas additionally intensifies the scarcity of Poland's water resources. These resources are, on average, three times lower than in other EU countries. Montane areas satisfy Poland's water requirements in 35%. In addition, montane areas are a source of mineral and medicinal waters. Therefore, the hydrological role of montane areas should be regarded as strategic, and these lands require special treatment. The hydrological importance of montane areas is additionally increased by the fact that Poland is a poor country in terms of water resources.

Low total precipitation as well as predominance of light soils with poor water retention and storage are the main reasons why water resources in Poland are so scarce. Many authors [1–6] show that, additionally, the fluctuations of water resources are significantly dependent on the way the catchment area is used.

A special role in the quantitative and qualitative fluctuations of water resources is attributed to grassland. Grassland is regarded as a natural biological filter. Turf protects the soil against erosion and slows down the outflow of meteoric water, changing surface runoff into subsurface runoff. Additionally, turf retains considerable amounts of biogenic elements and protects soil against being depleted of minerals.

In montane areas in Poland, meadows are predominant in some regions, and in others, they constitute a considerable percentage of agricultural land. It is 62.3% in the

montane zone of the Carpathians and 39.7% in the submontane zone. Permanent grassland constitutes 32.9% and 27.8% of agricultural land in the Sudetes and in the Świętokrzyskie Mountains, respectively.

Montane areas in Poland are in three geological formations: the Carpathians, the Sudetes, and the Świętokrzyskie Mountains. The Carpathians are the largest mountain range [7], with surface area of approximately 19,600 km$^2$. This constitutes 6% of the country's surface area. Montane areas are perceived as water-bearing. This is because they satisfy 30–35% of water requirements of the country [8,9]. As much as 13% of Poland's water resources fall on the Carpathians [10]. Hence, montane areas have strategic importance, as they ensure country's hydrological safety.

Poland is a country with scarce water resources, which are assessed based on the amount of runoff waters [11]. This strengthens the hydrological role of montane areas. The amount of water per Polish citizen is nearly three times smaller than the average for EU countries [12]. This results from the fact that Poland's water is merely 2.1% of the EU's water resources, and Polish citizens constitute approximately 5.5% of the Commonwealth's population [13]. The issue of insufficient access to clean water may affect many countries in future and become one of the greatest developmental challenges for mankind [5].

Fluctuations of water resources in quantitative and in quantitative terms depend on land use, relief, type of flora, intensity of agricultural production, firmness of soils, level of precipitation and locations of settlement units [14,15]. The reasons for Poland's scarce water resources should be sought mainly in the relatively low annual total precipitation. However, the water retention capacity is also limited by a large percentage of light soils (which have low retention and a low storage coefficient) [16].

Hydrological resources of montane and submontane areas are shaped mainly by the substantial amount of precipitation and a considerable runoff of relatively high-quality waters [17]. However, adapting the manner of soil use to natural conditions may reduce the negative impact that the manner of management has on the natural environment. The last three decades have witnessed an increase in the share of forested and turf-covered areas in Poland (mainly at the cost of arable land), but also an increase in the area of urbanized regions [18,19]. The increase in forested and turf-covered area should be regarded as beneficial for the protection of water resources, as opposed to the increase in the area urbanized regions.

Rural areas are greatly responsible for water pollution. Duer et al. [20] as well as Mioduszewski [21] state that approximately 50–60% of nitrogen flowing from Poland to the Baltic Sea comes from agriculture. Kristensen et al. [22], Kangas and Sanna [23], as well as Kazutaka et al. [24] highlight that ammonia emissions during storage of natural fertilizers and fertilizing play a considerable part in the contamination of the soil environment and water. Ammonia is also one of the sources of acid precipitation. Bogdał and Ostrowski [25] draw attention to the impact the agricultural character of land use has on the quality and chemical composition of water running off a submontane catchment. When comparing the composition of meteoric water to runoff water, the authors showed that water running off the catchment had three-, two-, and four-fold higher concentrations of N-NO$_3$, PO$_4$, and SO$_4$, respectively, compared to meteoric water. Sapek [26] indicates that not only industry and agriculture are responsible for the presence of biogens in the environment (including water). People living in rural areas with unregulated water and sewage management play a considerable part in the proliferation of contaminants. The author also points out that large quantities of biogenic elements enter the environment as a result of inefficient use of food that ends up in trash cans.

The studies of Twardy and Kopacz [27,28] as well as Twardy et al. [6] provide evidence for the positive effect of turf-covered and forested areas on the water environment. These studies show that the load of minerals leached from the less forested and turf-covered catchment was 10% larger than the load leached from the more forested and turf-covered catchment. When studying water quality in five rivers with different catchments, Skorbiłowicz [3] showed that river water in catchments with predominant forests and meadow

communities had the lowest nitrogen concentration. Based on modeling studies, Śmietanka [5] showed that a 10% increase in the acreage of grassland at the cost of arable land reduced the catchment load of nitrogen by 20%. A 20% increase in grassland area reduced the catchment load of nitrogen by 40%. Similarly, when assessing the effect of using three types of catchment on the quality of water in bodies of water, Kornaś and Grześkowiak [2] showed that the greatest quantities of nitrogen and phosphorus compounds flowed out of the catchment that had a large share of agricultural land and a low share of forest area. The amount of leached nutrients was inversely proportional to the share of forest communities in the catchment area. Mioduszewski [21] supports the thesis that grassy ecosystems ensure rational use of nature's resources because they reduce many destructive processes, namely they reduce soil erosion as well as runoff of water and nutrients but increase the biological activity. Traczyk [29] observes that grassy ecosystems play an important role in the purification of the natural environment. The author states that meadow vegetation accumulates approximately 980 kg of macroelements (N, P, K, Ca, Mg, Na) in biomass produced on 1 ha, whereas cereals accumulate only about $\frac{1}{4}$ of this quantity.

Grasslands predominate in the structure of agricultural lands in montane areas in Poland [30]. These lands generally have a fodder function, providing fodder for ruminants. These areas also play a special role in the quantitative and qualitative fluctuations of water resources [31,32]. Grasslands protect the soil against erosion and slow down the runoff of meteoric water, changing surface runoff into subsurface runoff and retaining considerable quantities of biogenic elements. For this reason, they constitute a natural biological filter that slows down the outflow of nutrients from the soil. At the same time, this factor decides soil fertility and the quality of moving water [33,34]. The manner of use and fertilization of grassland in montane and submontane areas has a significant impact on the quantity and quality of water resources. Hence, there is a strong relationship between the fodder function and the hydrological function of these areas [35].

Burzyńska [36] showed that meadow vegetation significantly reduces leaching of organic and inorganic nutrients from the soil to ground waters. Nutrient retention in the soil covered with grasses results from the formation of complexes through the binding of minerals with dissolved organic-mineral forms of carbon. According to the results of the cited study, leaching of phosphorus and magnesium was increasing, and leaching of manganese and zinc was decreasing along with decreasing soil acidity. Apart from environmental damage, incomplete fertilizer use by plants also causes financial losses for farmers [37]. However, some studies do not confirm that reduction of fertilization reduces the concentration of biogenes in leachate water. Jaguś [34] indicates that reducing the doses of mineral fertilizers reduced the concentration of nitrate ions in leachate water and did not have significant effect on the concentration of ammonium ions. The study suggests that natural fertilizers are a significant source of nitrogen compounds penetrating (with water) deep into the soil. Kopeć [38] does not confirm the opinion that organic fertilization has a significant impact on the contamination of runoff water. In this author's study, the amounts of nitrogen leached from the plots fertilized with manure (regardless of the date of application) were similar to those from the non-fertilized plot.

When analyzing the quality of drainage water flowing out of a meadow located in submontane areas, Jancovic i Folkman [39] established that the chemical composition of the water depended on the type of vegetation cover, number of species, root thickness, and defoliation frequency. Jaguś and Twardy [40] studied the effect of frequency of use of a grassy community on the amount and chemical composition of leachate moving through the soil profile. These authors found the lowest runoffs in the case of less frequent use (with one or two mowings of sward in the growing season). With increasing frequency of use, an increase in the amount of subsurface runoff and in the amount of leached nutrients (with the exception of phosphorus) was recorded. The study proved that the amount of water moving through the soil profile and the amount of macroelements leached with it were 1.5-fold higher in areas with no ground cover than in turf-covered areas. Misztal [41] also observed an increased (1.5-fold) water runoff from areas without ground cover (black

fallow) in relation to meadow. However, the author observes that water runoff from the soil in which oat was grown was lower than that from the turf area. This resulted from higher evapotranspiration from the soil in which oat was grown. Szajda and Łabędzki [42] recorded a negative correlation between the amount of water leaching through the soil profile and the magnitude of yield of grasses. Jankowski et al. [43] noticed that the longer the period for which montane areas are not used for feed functions, the less significant their role as a natural biological filter.

Benson et al. [44] notice that the amount of leachate water and the amount of nutrients leached with it is, aside from ground cover, also affected by the type of soil. Larger quantities of nutrients are leached from soils with low water capacity and low capillary rise. Filipek and Kasperczyk [45] recorded an effect of increased fertilization (particularly with nitrogen) on loosening of the turf. This leads to an increase in access of air to the soil and in the degree of mineralization of organic substance. Relationships between the aeration of the soil profile and the amount of released biogenic elements and their migration to water resources are also reported by Jaszczyński [46], Koc et al. [47], Paul and Clark [48], Sapek and Kalińska [49], Sapek et al. [50], Szajdak et al. [51], Skorbiłowicz [52], and Terlikowski [53]. Jaszczyński et al. [54] showed a seven times higher enrichment of underground water with nitrogen compounds at increased aeration of the soil profile (arising out of a decrease in the level of underground water). Kobuz [55] assumes that there is about 90% of soil nitrogen in the soil organic substance. Paul and Clark [48] assume that the degree of nitrogen use by plants, and at the same time the level of its leaching from the soil, depend on the degree of mineralization of organic matter in the soil. Terlikowski [53] showed that the amount of nitrogen released as a result of mineralization in the soil may cover about 40% of plant requirements for this nutrient. This author suggests that the most rational use of this element by meadow vegetation occurs under conditions of two to three meadow mowings.

Studies by Krawczyk et al. [56] and Pietrzak [57] are in certain opposition to the opinions that support the thesis stating that grassy communities play a positive role in reducing the leaching of biogenes from the soil. When comparing the impact of natural fertilizers on the soil environment and on the water environment of arable land and grassy communities in lowland and submontane areas, Krawczyk et al. [56] showed that, in spring, nitrogen content in the soil filtrate on arable land was four times higher and phosphorus content was two times higher than in runoff from turf-covered areas. In autumn, these authors observed a reverse relationship—the filtrate from the turf community was generally twice more abundant with the mentioned elements than the filtrate from arable land. The authors do not explain the reasons for the interseasonal variation of nutrient leaching. Presumably, such variation should be linked to higher supply and mineralization of organic substance in turf-covered areas. Pietrzak [57] comes to conclusions that challenge the role of grassland in water protection. When assessing nitrogen losses in farms with different percentages of grassland in the structure of agricultural lands, this author established that excess nitrogen in a farm is positively correlated with the percentage of grassland. Studies tackling the issue of nitrogen balance in farms with different production specialization that were carried out in France [58] indicate that the greatest excess nitrogen occurs in farms specializing in breeding slaughter cattle.

The decrease in the number of ruminants that can be observed in montane areas results in spreading of weeds at the cost of grasses and leads to thinning of turf. Sapek and Kalińska [49] indicate that abandoning the use of grassy areas also becomes a cause of water environment contamination. This results from the fact that abandoning mowing or grazing leads to accumulation of nitrogen compounds in the soil, which favors nitrification and leaching of nitrates. Kopeć [38] established that the load of nitrogen leached from the soil in sheepfold plots was similar to the amount leached from plots where a double dose of mineral nitrogen had been applied. The impact of the type of use (grazing) on the quality of runoff water was also observed by Kacorzyk et al. [59]. These authors established that the loads of nitrogen, phosphorus, and potassium that leached from the plot with a tight fold

over three years were almost twice as high as from the plot where mineral fertilization with similar doses of phosphorus and potassium and with a twice higher dose of nitrogen had been applied. The authors explain that the substantial loads of the fertilizer constituents leached from plots with folds by the stimulating effect of fresh animal excreta on leaching of nutrients from soil reserves. However, when assessing the effect of using swards differing in plant composition for grazing and mowing, Wasilewski and Sutkowska [60] observed that with lower loading of pastures, the amounts of nitrogen and potassium leached into underground water were similar to those from mowed meadow, constituting 5% of the fertilization dose. When the pasture loading was higher, leaching reached 10% of the fertilization dose and was many times higher than from the non-fertilized object.

Areas that are primary sites where water resources are generated should have an organized and a fully functional environmental protection system. Planning the development of these areas should include such forms of management that lead to the improvement of quality and stabilization of water runoff to watercourses and water reservoirs. Water resources should be shaped at all stages of water circulation, starting with the selection of the type of soil vegetation cover, through its retention, to the quantity and quality of water flowing out from the catchment [61].

There is relatively little research on the effect of natural fertilizers and the type of plants cultivated in montane and submontane areas on quantitative and qualitative fluctuations of water resources. Due to a decreasing number of ruminants, considerable grassland areas are being excluded from use. This raises concerns about the negative effect of abandoning the use of grass communities on fluctuations and quality of water resources. Under conditions of limited water resources and considerable diversification in their distribution over time (drought, heavy downpours), learning about the relations between the character of economic use of montane areas and the amount and quality of water resources enables conscious shaping of the relationship between economic development and the natural environment. Knowledge of this field enables one to shape such a ground cover that will have a beneficial effect on the quantitative and qualitative water reserves of the catchment.

The aim of the study was to assess the effect of ground cover on the quantitative and qualitative fluctuations of water resources in the submontane region. In particular, the following were assessed: the amount and chemical composition of leachate water, as well as the magnitude of loads of minerals leached with it in three research periods. The research process was carried out according to the scheme shown in Figure 1.

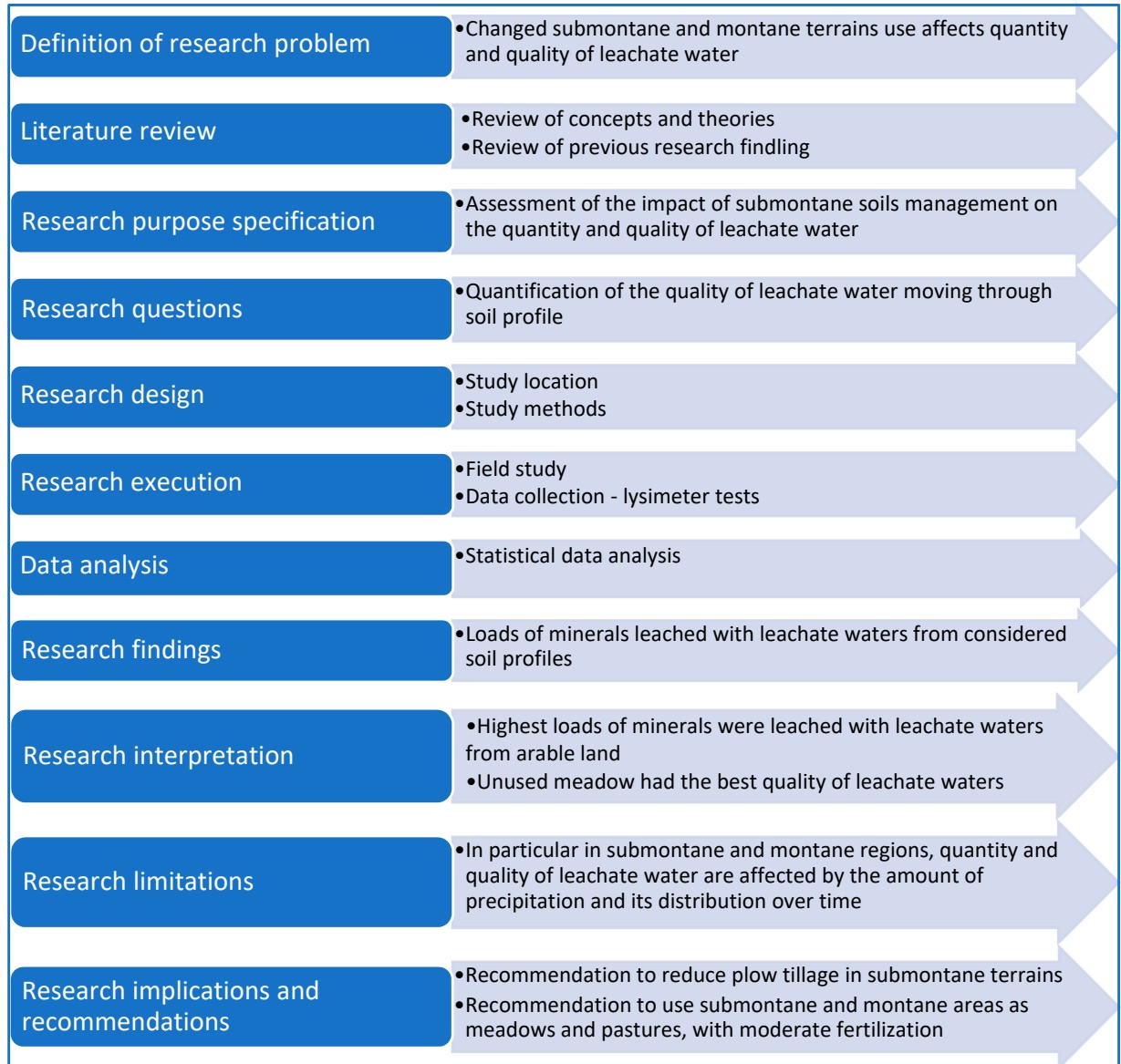

**Figure 1.** Flow chart of the research process.

## 2. Materials and Methods

### 2.1. Location of the Study

The study was carried out in Poland, in the eastern part of the Małopolska Province, which is a mountain range in the Western Carpathians. Soils occurring in this area are typical submontane and montane formations: brown acidic and leached, argillaceous and skeletal. In terms of agricultural suitability, most soils are classified in the IV and V soil quality classes. Only about 2% of soils can be classified as class II. The growing season in this region lasts 150–180 days, and the time of snow cover deposition is approximately 150 days.

The experiment was set up in the Wiśnicz Piedmont, at the Uszwica River at a height of 360 m above sea level, on a slope with a 3-degree gradient towards south-west. The study was carried out in the years 2016–2018. Brown acidic soil, with the granulometric composition of medium skeletal clay, was present in the experimental field. Physicochemical properties of the soil were as follows: pH in 1 mol·dm$^{-3}$ KCl was 3.2; organic matter content was 2.8%; total N content was 0.19%; content of available forms of P, K, and Mg

was 7.0, 42.0, and 131.5 mg·kg$^{-1}$ of dry matter, respectively. Maximum water capacity determined with intact soil structure was 59%, and field water capacity was 29%.

### 2.2. Study Methods

The experiment was set up in a randomized block design in three replications in 2015. Within the premises of the experiment, two areas differing in soil vegetation cover were sectioned off: permanent meadow with predominant share of velvet grass (*Holcus lanatus* L.), and arable land. The meadow region where experimental variants were sectioned off had been used as pasture for two decades. The area of arable land was allocated at random in each block and was created by plowing the grassland in 2015. Within an experimental plot, the following fertilization variants were considered:

(A)　non-fertilized meadow—-control;
(B)　meadow fertilized with mineral fertilizers ($P_{18}K_{50}N_{120}$);
(C)　meadow fertilized with 15 Mg·ha$^{-1}$ liquid manure + supplementary mineral fertilization ($P_{12}N_{45}$);
(D)　meadow fertilized with 10 Mg·ha$^{-1}$ manure + supplementary mineral fertilization ($P_4N_{51}$);
(E)　barren meadow (non-fertilized, non-mowed);
(F)　arable land receiving mineral fertilizers in the amount of $P_{18}K_{50}N_{120}$ for the first two years and $P_{18}K_{50}N_{60}$ in the third year of use.

Barley was grown on the arable land in 2016, in the next year, oat, and after its harvest, a mixture of Poaceae and legumes: perennial ryegrass (*Lolium perenne* L.) was sown in August. A total of 80% + red clover (*Trifolium pratense* L.) 20%, in the amount of 40 kg·ha$^{-1}$, constituted the main yield in the following year. All cultivation measures (pre-sowing, post-sowing) were performed manually due to the installed lysimeters.

Fertilization was applied in each year of the experiment. Nitrogen in the form of ammonium nitrate (34% N) in plot B was applied in two parts: 60% for the first regrowth and 40% for the second regrowth. In other objects with mineral fertilization, ammonium nitrate was applied once in spring. Phosphorus in the form of superphosphate (40% $P_2O_5$) and potassium in the form of potassium salt (56% $K_2O$) were applied in their entirety at the beginning of the growing season. Natural-mineral fertilization was selected in such a way that, after accommodating the content of fertilizer components included in manure and liquid manure, the amount of supplied P and N was the same as in plot B (Table 1). Liquid manure was used in two parts: a 60% dose under the first regrowth, and the remaining part under the second regrowth. Manure was applied in early spring before the growing season. In 2018, on the mowed meadow plots (A, B, C, D) as well as on the arable plot (F), the first regrowth was harvested at the beginning of flowering of meadow fescue (*Festuca pratensis* Huds.) in June, and the second regrowth in the third decade of August. On arable land (F) in 2016 and 2017, the harvest was conducted once the grains had reached full maturity. Biomass was harvested from the area of 15 m$^2$ of each plot, except from object E (non-fertilized and non-mowed meadow).

**Table 1.** Chemical composition of organic fertilizers.

| Nutrient | Sheep Manure | Liquid Manure |
|---|---|---|
| | (g kg$^{-1}$ Fresh Matter) | |
| N | 6.9 | 5.0 |
| P | 1.4 | 0.3 |
| K | 6.0 | 7.0 |
| Ca | 2.5 | 0.4 |
| Mg | 0.8 | 0.5 |
| Na | 0.6 | 3.4 |

Samples of natural fertilizers collected for chemical analysis in individual years of the study came from ongoing production and differed from one another slightly. Manure contained 23.2–23.7% dry matter, and liquid manure contained 6.3–6.8% dry matter.

There were 3 lysimeters modified by Szyłowa in each variant, placed at the depth of 0–30 cm. The lysimeters were mounted in 2015. They were used for the assessment of leachate water that was later subjected to chemical analyses. The cumulative area of each lysimeter (circular, with a diameter of 50 cm) was 1962.5 cm$^2$. Bottoms of lysimeters had funnel-shaped ends filled with a small amount of gravel that acted as a filter protecting the discharge pipes against clogging. The funnel-shaped ends of lysimeters had tubes attached, through which leachate water flowed into tanks located outside the area of the experiment in a specially prepared basement. Precipitation was measured using the wireless Davis Vantage Pro 2 (6152EU) weather station that was installed on the experimental field.

Each time after heavier precipitation, the volume of leachate water was measured and samples were collected for chemical analyses. The amounts of meteoric water and leachate water as well as their composition were determined in three seasons of each year of the study:

- first (I)—-between 1 April and 30 June 2015 (intensive growing season),
- second (II)—-between 1 July and 31 October 2015 (slow growing season),
- third (III)—-between 1 November and 31 March 2015 (non-growing season).

The above classification into seasons result from the necessity to determine what impact plant growing (or lack thereof) has on the quantity and quality of leachate water. Based on the magnitude of precipitation and after understanding the above relationships, one will be able to estimate the risk of flooding in the catchment.

Natural fertilizers intended for chemical analyses were subjected to dry mineralization, and ash was digested in H-NO$_3$ (1:3). The content of phosphorus, potassium, calcium, magnesium, and sodium was determined in the obtained solutions and in the leachate water using an inductively coupled plasma optical emission spectrophotometer (ICP-OES) manufactured by Perkin-Elmer, model Optima 7300 DV [62]. The above-mentioned analyses were carried out at the University of Agriculture. The content of mineral forms of nitrogen N-NO$_3$ + N-NH$_4$ in the water was determined using an LF-205 microprocessor photometer manufactured by Slandi. As there was no possibility of performing chemical analyses at the water sample collection site, the analyses were carried out in a laboratory several to a dozen or so hours after they had been collected. For this reason, among other things, trace amounts of ammonium nitrogen in water were determined. That is why the tables include the total content of both forms of nitrogen. Nitrogen content in the soil and in natural fertilizers was determined by Kjeldahl method on a KjelFlex K-360 apparatus [63]. The content of available forms of phosphorus and potassium in the soil was determined by the Egner–Riehm method, and the content of available magnesium by the Schachtschabel method [63]. The soil pH in water suspension and in 1 mol·dm$^{-3}$ KCl was determined by the potentiometric method [63].

The amount of leachate water, nutrient concentration in the water, and load of nutrients leached from 1 hectare were determined in each period of the study (I, II, and III). The mentioned values are shown in tables as arithmetic means from three years of the study. The loads of nutrients leached from the soil per 1 hectare were calculated by multiplying the amount of leachate water in the period of study in a given year from 1 hectare by the concentration of a given nutrient in that water. Afterwards, the arithmetic mean was calculated for a given period within the three years of the study. Changes in the concentration of N-NO$_3$ + N-NH$_4$, P, K, Ca, Mg, and Na (in mg·dm$^{-3}$) in soil leachates are presented by calculating the standard deviation (SD) and variation coefficients (V%).

In studying the relationships between the loads leached from the soil and the previously identified study variants and periods, recurrent methods of partitioning of a data set into disjoint subsets were used—classification and regression tree analysis (C&RT). The results of this hierarchical procedure of discriminating a set of objects are presented in Figures A1–A6.

Basing a model for describing the fluctuations of the magnitude of nutrient loads leached with leachate waters from studied variants in individual periods on a nonparametric method does not involve the necessity to reveal predictors in the early stages of a study, because the role of variables is defined in the course of model formulation. Using this nonparametric method does not exempt one from the necessity to know the breakdown of features or from meeting restrictive assumptions within this respect either. A description of a method of discrimination of data sets using the classification trees method can be sought in numerous elaborations on this subject, among others in studies by Džeroski and Lavrač [64], Jones [65], Navulur [66], Rao at all. [67], and Rokach and Maimon [68].

## 3. Results

The quality of water resources and soil depletion of minerals depend on the magnitude of water runoff from topsoil as well as on the concentration of nitrogen, phosphorus, potassium, calcium, magnesium, and sodium in that water.

### 3.1. Nitrogen

In most objects, leachate water had the lowest concentration of nitrogen in $N-NO_3$ + $N-NH_4$ forms in the first period of the study, i.e., during intensive growing, and the highest concentration in the third period (non-growing season) (Table 2).

**Table 2.** Concentration and load of nitrogen ($N–NO_3$ + $N–NH_4$) diluted from 0–30 cm soil layer with migrated rain water.

| Variant | Period | Concentration of N (mg·dm$^{-3}$) | SD | V% | 1 | 2 | 3 | Mean |
|---|---|---|---|---|---|---|---|---|
| | | | | | \multicolumn: Year of the Research | | | |
| | | | | | \multicolumn: Load of N (kg·ha$^{-1}$) | | | |
| Control | I | 0.74 | 0.42 | 56.97 | 0.39 | 0.27 | 0.02 | 0.22 |
| | II | 2.05 | 0.72 | 34.90 | 1.61 | 1.69 | 2.12 | 1.81 |
| | III | 2.73 | 0.58 | 21.23 | 1.72 | 0.50 | 0.45 | 0.89 |
| | Σ | - | - | - | 3.72 | 2.46 | 2.59 | 2.92 |
| Mineral P$_{18}$K$_{50}$N$_{120}$ | I | 1.37 | 0.38 | 27.70 | 0.67 | 0.58 | 0.05 | 0.43 |
| | II | 3.10 | 1.44 | 46.28 | 2.27 | 5.86 | 1.95 | 3.36 |
| | III | 5.36 | 1.77 | 32.92 | 2.68 | 1.79 | 0.66 | 1.71 |
| | Σ | - | - | - | 5.62 | 8.23 | 2.66 | 5.50 |
| Liquid manure 15 Mg + P$_{14}$N$_{45}$ | I | 0.90 | 0.26 | 29.40 | 0.39 | 0.33 | 0.03 | 0.25 |
| | II | 5.32 | 2.64 | 49.66 | 2.55 | 9.46 | 5.71 | 5.91 |
| | III | 3.47 | 1.02 | 29.46 | 2.62 | 1.05 | 0.34 | 1.33 |
| | Σ | - | - | - | 5.55 | 10.84 | 6.08 | 7.49 |
| FYM (spring) 10 Mg + P$_4$N$_{51}$ | I | 0.81 | 0.24 | 29.89 | 0.33 | 0.39 | 0.02 | 0.25 |
| | II | 2.48 | 0.64 | 25.70 | 2.25 | 4.52 | 2.74 | 3.17 |
| | III | 4.94 | 1.96 | 39.78 | 4.46 | 0.90 | 0.50 | 1.95 |
| | Σ | - | - | - | 7.05 | 5.81 | 3.26 | 5.37 |
| Fallow | I | 0.34 | 0.14 | 40.00 | 0.20 | 0.19 | 0.01 | 0.13 |
| | II | 2.05 | 0.76 | 37.15 | 2.24 | 1.61 | 2.38 | 2.08 |
| | III | 4.34 | 1.08 | 25.00 | 2.69 | 1.25 | 0.44 | 1.46 |
| | Σ | - | - | - | 5.14 | 3.04 | 2.83 | 3.67 |
| Arable land P$_{18}$K$_{50}$N$_{60-120}$ | I | 0.85 | 0.61 | 72.29 | 0.59 | 0.31 | 0.02 | 0.30 |
| | II | 10.20 | 3.46 | 33.96 | 18.13 | 12.43 | 8.36 | 12.97 |
| | III | 5.67 | 1.50 | 26.49 | 5.27 | 1.52 | 0.67 | 2.49 |
| | Σ | - | - | - | 23.99 | 14.26 | 9.05 | 15.76 |

In the first period, nitrogen concentration varied from 0.34 mg·dm$^{-3}$ in the object with barren meadow to the 1.37 mg·dm$^{-3}$ in the object receiving mineral fertilization. In the third period, i.e., the non-growing season, nitrogen concentration in leachate water ranged from 2.73 to 5.67 mg·dm$^{-3}$. Difference in nitrogen concentration between these periods was even five- to six-fold. The exception was water of two objects: fertilized with liquid

manure and arable land. In those objects, water in the second period contained the greatest amount of nitrogen. When analyzing the magnitudes of these loads leached in individual periods, it was established that the largest loads were leached in the second, and the lowest in the first period of the study. Hence, the period of study emerges as the most important criterion diversifying the leaching of nitrogen loads from the soil between the investigated variants of use of submontane areas (Figure A1).

In the objects receiving mineral fertilization and liquid manure, the largest loads of this nutrient were leached from the soil in the second year of the study. In other objects, these loads were highest in the first year of the study. On average, the highest annual load of nitrogen was leached from arable land. It was two- to three-fold higher than the loads leached from the other objects.

### 3.2. Phosphorus

Leachate water in most objects had the lowest concentration of phosphorus in the first period (intensive growing), and the highest in the second period (slow growing) (Table 3).

**Table 3.** Concentration and load of phosphorus (P) diluted from 0–30 cm soil layer with migrated rain water.

| Variant | Period | Concentration of P (mg·dm$^{-3}$) | SD | V% | Year of the Research | | | Mean |
|---|---|---|---|---|---|---|---|---|
| | | | | | 1 | 2 | 3 | |
| | | | | | Load of P (kg·ha$^{-1}$) | | | |
| Control | I | 0.36 | 0.11 | 30.50 | 0.10 | 0.12 | 0.02 | 0.08 |
| | II | 0.36 | 0.23 | 63.09 | 0.09 | 0.49 | 0.38 | 0.32 |
| | III | 0.45 | 0.00 | 0.25 | 0.28 | 0.11 | 0.06 | 0.15 |
| | Σ | - | - | - | 0.47 | 0.72 | 0.47 | 0.55 |
| Mineral P$_{18}$K$_{50}$N$_{120}$ | I | 0.46 | 0.07 | 15.36 | 0.15 | 0.26 | 0.02 | 0.14 |
| | II | 0.41 | 0.27 | 67.58 | 0.31 | 0.25 | 0.59 | 0.38 |
| | III | 0.65 | 0.43 | 66.04 | 0.59 | 0.22 | 0.02 | 0.28 |
| | Σ | - | - | - | 1.05 | 0.72 | 0.64 | 0.80 |
| Liquid manure 15 Mg + P$_{14}$N$_{45}$ | I | 0.33 | 0.05 | 15.91 | 0.13 | 0.13 | 0.01 | 0.09 |
| | II | 0.43 | 0.26 | 60.75 | 0.77 | 0.24 | 0.36 | 0.46 |
| | III | 0.33 | 0.32 | 96.43 | 0.47 | 0.03 | 0.02 | 0.18 |
| | Σ | - | - | - | 1.37 | 0.40 | 0.40 | 0.72 |
| FYM (spring) 10 Mg + P$_4$N$_{51}$ | I | 0.32 | 0.07 | 22.73 | 0.13 | 0.12 | 0.01 | 0.09 |
| | II | 0.34 | 0.04 | 12.76 | 0.46 | 0.44 | 0.38 | 0.43 |
| | III | 0.47 | 0.30 | 64.54 | 0.29 | 0.18 | 0.02 | 0.16 |
| | Σ | - | - | - | 0.88 | 0.74 | 0.41 | 0.68 |
| Fallow | I | 0.27 | 0.17 | 61.25 | 0.05 | 0.15 | 0.02 | 0.07 |
| | II | 0.51 | 0.13 | 24.65 | 0.54 | 0.49 | 0.54 | 0.52 |
| | III | 0.26 | 0.06 | 21.31 | 0.20 | 0.05 | 0.03 | 0.09 |
| | Σ | - | - | - | 0.79 | 0.68 | 0.60 | 0.69 |
| Arable land P$_{18}$K$_{50}$N$_{60-120}$ | I | 0.58 | 0.41 | 70.88 | 0.17 | 0.12 | 0.05 | 0.12 |
| | II | 1.00 | 0.82 | 81.96 | 0.38 | 1.21 | 1.94 | 1.18 |
| | III | 0.81 | 0.55 | 68.16 | 0.88 | 0.29 | 0.03 | 0.40 |
| | Σ | - | - | - | 1.43 | 1.62 | 2.02 | 1.69 |

The exception was water in objects receiving mineral fertilization and manure. In those objects, the lowest concentration of phosphorus in water was recorded in the third period, i.e., the non-growing season. On average, the lowest load of phosphorus was leached (with leachate water) from the soil in the first period of the study, and the highest in the second period. Total annual load of leached phosphorus was generally highest in the first year, and in arable land in the third year. On average, annual loads of phosphorus leached from arable land were twice higher than the loads leached from meadow objects.

The period of the study becomes the most important factor that determines phosphorus leaching from the soil for the considered variants of land use (Figure A2). The highest

loads of phosphorus in leachate water were recorded in the second period in arable land and in the object fertilized with liquid manure. In the other periods, the meadow object with mineral fertilization had higher leaching of phosphorus than the other variants of the study.

### 3.3. Potassium

In general, leachate water had the lowest potassium concentration in the third period (non-growing season) and the highest in the second period (slow growing season) (Table 4).

**Table 4.** Concentration and load of potassium (K) diluted from 0–30 cm soil layer with migrated rain water.

| Variant | Period | Concentration of K (mg·dm$^{-3}$) | SD | V% | Year of the Research | | | Mean |
|---|---|---|---|---|---|---|---|---|
| | | | | | 1 | 2 | 3 | |
| | | | | | Load of K (kg·ha$^{-1}$) | | | |
| Control | I | 1.85 | 0.42 | 22.57 | 0.75 | 0.76 | 0.06 | 0.52 |
| | II | 1.39 | 0.09 | 6.47 | 1.37 | 1.53 | 0.96 | 1.29 |
| | III | 1.12 | 0.08 | 6.79 | 0.71 | 0.25 | 0.16 | 0.37 |
| | Σ | - | - | - | 2.82 | 2.54 | 1.18 | 2.18 |
| Mineral P$_{18}$K$_{50}$N$_{120}$ | I | 2.24 | 0.43 | 19.03 | 0.70 | 1.01 | 0.13 | 0.61 |
| | II | 2.27 | 1.15 | 50.52 | 2.18 | 1.50 | 2.90 | 2.19 |
| | III | 2.17 | 0.23 | 10.68 | 1.42 | 0.58 | 0.28 | 0.76 |
| | Σ | - | - | - | 4.30 | 3.09 | 3.30 | 3.56 |
| Liquid manure 15 Mg + P$_{14}$N$_{45}$ | I | 1.11 | 0.71 | 63.79 | 0.18 | 0.36 | 0.08 | 0.21 |
| | II | 2.13 | 0.64 | 29.95 | 2.86 | 1.84 | 2.13 | 2.28 |
| | III | 1.14 | 0.26 | 23.24 | 0.76 | 0.22 | 0.20 | 0.39 |
| | Σ | - | - | - | 3.80 | 2.41 | 2.41 | 2.87 |
| FYM (spring) 10 Mg + P$_4$N$_{51}$ | I | 1.07 | 0.50 | 47.01 | 0.20 | 0.45 | 0.07 | 0.24 |
| | II | 1.96 | 0.87 | 44.17 | 3.56 | 1.61 | 2.07 | 2.41 |
| | III | 2.84 | 1.47 | 51.78 | 2.38 | 0.81 | 0.16 | 1.12 |
| | Σ | - | - | - | 6.14 | 2.87 | 2.30 | 3.77 |
| Fallow | I | 1.97 | 0.66 | 33.44 | 0.52 | 1.32 | 0.13 | 0.66 |
| | II | 2.46 | 0.40 | 16.13 | 2.14 | 3.45 | 2.26 | 2.62 |
| | III | 1.30 | 0.60 | 46.17 | 0.69 | 0.19 | 0.27 | 0.38 |
| | Σ | - | - | - | 3.34 | 4.97 | 2.65 | 3.65 |
| Arable land P$_{18}$K$_{50}$N$_{60-120}$ | I | 2.71 | 1.84 | 67.76 | 1.84 | 0.88 | 0.08 | 0.93 |
| | II | 2.35 | 0.16 | 6.82 | 2.99 | 3.82 | 2.24 | 3.02 |
| | III | 1.76 | 0.90 | 51.14 | 1.29 | 0.72 | 0.14 | 0.72 |
| | Σ | - | - | - | 6.12 | 5.42 | 2.46 | 4.67 |

The exception was water from the object fertilized with manure where the greatest amount of potassium was recorded in the third period. Water of the minerally fertilized object deserves attention because samples in all periods had a similar potassium concentration. The highest potassium load was leached with water from the soil in the second period of the study. With regard to potassium, the time of sample collection also appears to be the most important factor determining the amount of leached nutrient (Figure A3). In the second period, the relatively highest loads of potassium were leached from arable land and from the object fertilized with liquid manure. In the other periods (particularly in the third one), the highest loads of potassium were leached from areas supplied with mineral fertilizers, and the lowest from barren meadow.

In the utilized objects, the highest loads of potassium from the soil were leached in the first year of the study, and in barren meadow in the second year. On average, the highest annual leaching of potassium was recorded in arable land, and the lowest in the control. The difference between these values was over two-fold.



### 3.4. Calcium

Leachate water of most objects generally had a similar concentration of calcium in all periods of the study (Table 5).

**Table 5.** Concentration and load of calcium (Ca) diluted from 0–30 cm soil layer with migrated rain water.

| Variant | Period | Concentration of Ca ($mg \cdot dm^{-3}$) | SD | V% | Year of the Research | | | Mean |
|---|---|---|---|---|---|---|---|---|
| | | | | | 1 | 2 | 3 | |
| | | | | | Load of Ca ($kg \cdot ha^{-1}$) | | | |
| Control | I | 17.17 | 2.70 | 15.74 | 4.59 | 7.68 | 0.81 | 4.36 |
| | II | 16.65 | 5.06 | 30.39 | 11.17 | 17.26 | 16.28 | 14.90 |
| | III | 17.74 | 8.51 | 47.98 | 5.87 | 4.12 | 3.62 | 4.54 |
| | Σ | - | - | - | 21.63 | 29.06 | 20.71 | 23.80 |
| Mineral $P_{18}K_{50}N_{120}$ | I | 19.09 | 2.68 | 14.07 | 5.96 | 9.54 | 1.02 | 5.51 |
| | II | 22.98 | 5.39 | 23.46 | 20.85 | 24.19 | 24.18 | 23.08 |
| | III | 23.94 | 5.88 | 24.56 | 12.55 | 5.36 | 4.34 | 7.42 |
| | Σ | - | - | - | 39.37 | 39.09 | 29.55 | 36.00 |
| Liquid manure $15 Mg + P_{14}N_{45}$ | I | 10.48 | 1.58 | 15.08 | 3.79 | 3.63 | 0.45 | 2.62 |
| | II | 12.77 | 0.90 | 7.02 | 15.28 | 16.03 | 10.87 | 14.06 |
| | III | 11.30 | 1.77 | 15.63 | 7.58 | 3.25 | 1.39 | 4.08 |
| | Σ | - | - | - | 26.64 | 22.92 | 12.71 | 20.76 |
| FYM (spring) $10 Mg + P_4N_{51}$ | I | 12.76 | 2.58 | 20.26 | 3.48 | 5.36 | 0.67 | 3.17 |
| | II | 14.58 | 3.42 | 23.47 | 14.56 | 21.44 | 18.55 | 18.18 |
| | III | 19.77 | 6.11 | 30.91 | 15.12 | 5.08 | 1.73 | 7.31 |
| | Σ | - | - | - | 33.16 | 31.88 | 20.96 | 28.67 |
| Fallow | I | 12.41 | 1.33 | 10.75 | 4.66 | 7.16 | 0.74 | 4.19 |
| | II | 15.27 | 2.73 | 17.90 | 16.32 | 22.84 | 10.69 | 16.62 |
| | III | 16.51 | 3.93 | 23.80 | 7.44 | 4.42 | 2.47 | 4.78 |
| | Σ | - | - | - | 28.42 | 34.41 | 13.91 | 25.58 |
| Arable land $P_{18}K_{50}N_{60-120}$ | I | 27.14 | 16.28 | 60.00 | 16.89 | 5.89 | 1.24 | 8.01 |
| | II | 31.66 | 8.18 | 25.85 | 52.48 | 40.58 | 27.62 | 40.23 |
| | III | 18.63 | 5.58 | 29.93 | 13.63 | 3.54 | 3.86 | 7.01 |
| | Σ | - | - | - | 83.00 | 50.01 | 32.72 | 55.24 |

This concentration ranged from 10.48 to 23.94 $mg \cdot dm^{-3}$. Only in the object fertilized with manure was there an increase in calcium concentration in leachate water in consecutive periods of the study. When analyzing the magnitudes of leached calcium in individual periods, it was established that the largest loads of this nutrient were leached in the second, and the lowest in the first period. The highest loads of calcium were leached in the second year of the study, and the lowest in the third. On average, the highest annual load of calcium was leached from arable land. Its magnitude was 1.5 times higher than that of the load leached from the minerally fertilized object and over two times higher compared to the other objects.

The most important (of the studied) factors determining the load of calcium leached from the soil diverged little from analogous conditions for potassium (Figure A4). In the second period, the highest loads of calcium were leached from arable land and from the object supplied with mineral fertilizers. In the other periods, particularly high loads of calcium were leached from objects fertilized with organic materials. Particularly high loads of calcium were recorded in leachate water from arable land.

### 3.5. Magnesium

Leachate water from utilized meadow objects contained the largest amount of magnesium in the third period, and the lowest in the first period of the study (Table 6).

**Table 6.** Concentration and load of magnesium (Mg) diluted from 0–30 cm soil layer with migrated rain water.

| Variant | Period | Concentration of Mg (mg·dm$^{-3}$) | SD | V% | Year of the Research | | | Mean |
|---|---|---|---|---|---|---|---|---|
| | | | | | 1 | 2 | 3 | |
| | | | | | Load of Mg (kg·ha$^{-1}$) | | | |
| Control | I | 1.87 | 0.60 | 32.29 | 0.42 | 1.06 | 0.08 | 0.52 |
| | II | 2.51 | 0.36 | 14.36 | 2.31 | 3.15 | 1.58 | 2.35 |
| | III | 2.60 | 0.43 | 16.52 | 1.33 | 0.66 | 0.40 | 0.80 |
| | Σ | - | - | - | 4.06 | 4.86 | 2.06 | 3.66 |
| Mineral P$_{18}$K$_{50}$N$_{120}$ | I | 2.09 | 0.33 | 15.76 | 0.67 | 1.18 | 0.10 | 0.65 |
| | II | 2.23 | 1.08 | 48.48 | 3.39 | 1.37 | 1.92 | 2.23 |
| | III | 2.50 | 0.75 | 30.21 | 1.63 | 0.79 | 0.25 | 0.89 |
| | Σ | - | - | - | 5.69 | 3.33 | 2.27 | 3.76 |
| Liquid manure 15 Mg + P$_{14}$N$_{45}$ | I | 1.82 | 0.56 | 30.82 | 0.47 | 1.03 | 0.06 | 0.52 |
| | II | 1.80 | 0.18 | 9.80 | 1.87 | 2.63 | 1.50 | 2.00 |
| | III | 1.96 | 0.61 | 31.32 | 1.31 | 0.64 | 0.20 | 0.72 |
| | Σ | - | - | - | 3.66 | 4.29 | 1.76 | 3.24 |
| FYM (spring) 10 Mg + P$_4$N$_{51}$ | I | 1.76 | 0.69 | 39.47 | 0.39 | 0.69 | 0.11 | 0.40 |
| | II | 1.85 | 0.59 | 31.78 | 1.62 | 3.71 | 1.87 | 2.40 |
| | III | 3.90 | 1.00 | 25.50 | 3.13 | 0.80 | 0.43 | 1.45 |
| | Σ | - | - | - | 5.14 | 5.20 | 2.41 | 4.25 |
| Fallow | I | 1.54 | 0.20 | 13.13 | 0.56 | 0.93 | 0.09 | 0.53 |
| | II | 2.46 | 1.36 | 55.39 | 2.63 | 4.85 | 0.94 | 2.80 |
| | III | 1.48 | 0.61 | 41.29 | 0.78 | 0.23 | 0.29 | 0.43 |
| | Σ | - | - | - | 3.96 | 6.02 | 1.32 | 3.77 |
| Arable land P$_{18}$K$_{50}$N$_{60-120}$ | I | 3.38 | 0.79 | 23.50 | 1.57 | 1.25 | 0.17 | 1.00 |
| | II | 4.81 | 1.02 | 21.32 | 7.65 | 6.37 | 4.31 | 6.11 |
| | III | 2.69 | 0.06 | 2.21 | 1.96 | 0.71 | 0.44 | 1.04 |
| | Σ | - | - | - | 11.18 | 8.33 | 4.92 | 8.15 |

Water coming from arable land and barren meadow had the highest concentration in the second period, and the lowest in the third. The highest load of magnesium was leached in the second period of the study, whereas magnitudes of these loads in the first and third period were two- to three-fold lower than in the second period. In most objects, the highest load of magnesium was leached from the soil in the second year of the study. Water of the minerally fertilized object contained the largest amount of calcium in the first year. On average, the lowest annual load of magnesium was leached from the object fertilized with liquid manure, and the highest from arable land. In the latter case, this load was over two-fold higher than the loads from the other meadow objects.

Leaching of magnesium from the soil was differentiated mostly by the time of sample collection (Figure A5). In a tendency, the highest level of magnesium leaching was recorded in the second period of the study from arable land. A twice lower intensity of the process was observed in the barren meadow. In the other periods, the highest load of magnesium was leached from the object fertilized with manure.

### 3.6. Sodium

Leachate water from most objects had the highest concentration of sodium in the second period of the study (Table 7).

The minerally fertilized object and the object fertilized with manure were the exceptions to the above rule. The highest loads of sodium leached from the soil were recorded in the second period of the study. In that period, their magnitudes were four- to six-fold higher than in the first and third period. The largest amounts of sodium were generally leached in the second year of the study, and the lowest in the third. An exception was the minerally fertilized object, where the highest load of sodium was leached in the first year.

On average, the annual load of leached sodium in meadow objects was similar and ranged from 3.2 to 3.8 kg·ha$^{-1}$, whereas in arable land it was twice as high.

**Table 7.** Concentration and load of sodium (Na) diluted from 0–30 cm soil layer with migrated rain water.

| Variant | Period | Concentration of Na (mg·dm$^{-3}$) | SD | V% | Year of the Research | | | Mean |
|---|---|---|---|---|---|---|---|---|
| | | | | | 1 | 2 | 3 | |
| | | | | | Load of Na (kg·ha$^{-1}$) | | | |
| Control | I | 2.69 | 0.51 | 18.91 | 0.84 | 0.95 | 0.14 | 0.64 |
| | II | 2.73 | 0.85 | 31.05 | 1.83 | 2.81 | 2.69 | 2.44 |
| | III | 2.27 | 0.57 | 25.11 | 1.44 | 0.40 | 0.39 | 0.74 |
| | Σ | - | - | - | 4.10 | 4.15 | 3.22 | 3.82 |
| Mineral P$_{18}$K$_{50}$N$_{120}$ | I | 1.69 | 0.66 | 39.07 | 0.35 | 0.94 | 0.10 | 0.47 |
| | II | 1.98 | 0.54 | 27.17 | 2.50 | 1.70 | 1.78 | 1.99 |
| | III | 1.86 | 0.34 | 18.28 | 1.22 | 0.37 | 0.31 | 0.63 |
| | Σ | - | - | - | 4.06 | 3.01 | 2.20 | 3.09 |
| Liquid manure 15 Mg + P$_{14}$N$_{45}$ | I | 1.50 | 0.78 | 52.28 | 0.24 | 0.96 | 0.06 | 0.42 |
| | II | 1.74 | 0.34 | 19.50 | 1.58 | 2.76 | 1.48 | 1.94 |
| | III | 1.76 | 0.59 | 33.30 | 1.18 | 0.58 | 0.17 | 0.65 |
| | Σ | - | - | - | 3.01 | 4.31 | 1.72 | 3.01 |
| FYM (spring) 10 Mg + P$_4$N$_{51}$ | I | 2.24 | 0.93 | 41.35 | 1.09 | 0.89 | 0.06 | 0.68 |
| | II | 1.78 | 0.57 | 32.15 | 1.70 | 3.68 | 1.62 | 2.33 |
| | III | 2.18 | 0.93 | 42.53 | 1.34 | 0.72 | 0.17 | 0.74 |
| | Σ | - | - | - | 4.13 | 5.29 | 1.85 | 3.75 |
| Fallow | I | 1.74 | 0.94 | 54.27 | 0.33 | 1.48 | 0.10 | 0.64 |
| | II | 2.01 | 0.59 | 29.58 | 2.14 | 3.30 | 1.20 | 2.21 |
| | III | 1.37 | 0.67 | 49.32 | 0.53 | 0.26 | 0.29 | 0.36 |
| | Σ | - | - | - | 3.00 | 5.03 | 1.59 | 3.21 |
| Arable land P$_{18}$K$_{50}$N$_{60-120}$ | I | 2.12 | 1.12 | 52.60 | 0.37 | 1.10 | 0.16 | 0.54 |
| | II | 4.12 | 0.62 | 15.01 | 4.76 | 7.33 | 3.87 | 5.32 |
| | III | 2.78 | 0.21 | 7.42 | 2.03 | 0.81 | 0.41 | 1.08 |
| | Σ | - | - | - | 7.16 | 9.24 | 4.44 | 6.95 |

Similarly to other nutrients, sodium leaching from the soil was determined mostly by the time of measurement, reaching the highest intensification in the second period of the study (Figure A6). The greatest intensification of the process was recorded in the second period from arable land and from the control (a twice lower level than in arable land). In the other periods, sodium leaching from the control was only slightly higher than for the other study variants.

Despite the fact that the research was conducted over the course of three years, the obtained results reflect only general relations and trends in the observed processes. This is because the quantity and quality of leachate water are largely affected by the amount of precipitation and its distribution over time. This results from the fact that atmospheric precipitation and its distribution over time are extremely variable, particularly in submontane and montane regions. Therefore, lysimeter tests are extremely difficult and laborious.

The completed lysimeter tests were dynamic. The aim of the tests was to determine the amount and intensity of leached loads, as well as their characteristics. Owing to the complexity of the subject matter and surroundings of the tests, their results should not be interpreted in absolute terms in this case; they should rather be interpreted as certain recommendations.

## 4. Discussion

Of the obtained results, the following relationships need to be clarified:

- the lowest concentration of macroelements in leachate water in the first period of the study (intensive growing);
- differences in concentration of potassium, phosphorus, and calcium in leachate water between objects receiving mineral and natural fertilizers;
- the highest load of minerals leached with leachate water from the soil in the second period of the study (slow growing season);
- considerable differences in the magnitude of load of macroelements leached from the soil between turf-covered objects (meadow objects) and arable soil.

The index of water leaching from the soil was the lowest in the first period of the study. This enabled an assumption that concentration of nutrients in this small quantity of leachate water should be relatively high. However, this relationship was not confirmed. Two facts decided on such a low concentration of nutrients: intensive uptake of these nutrients by luxuriantly growing grassy vegetation as well as their relatively low supply in the soil on account of slow mineralization of organic substance by relatively low warming up of the soil. A study by Sapek and Kalińska [49] shows that the greatest dynamics of organic substance mineralization in soil in central Poland can be observed in the period from May to July. This study was located in a submontane area, where (on account of delayed growing) mineralization reaches its maximum later, namely in the period from June to August. This explains generally the highest concentration of nutrients in leachate water in the second period of the study. The higher concentration of potassium, phosphorus, and calcium in leachate water in the third period in objects receiving natural fertilizers should be explained by the fact that fertilization with natural fertilizers leads to immobilization of nutrients in the soil as a result of an increasing population of microorganisms that utilize these nutrients. Sapek and Kalińska [49], Wesołowski and Durkowski [69], and Mazur et al. [70] reached the same conclusions. In the study by Mazur et al. [70], after fertilization with manure, intensified nitrification lasted much longer than in objects fertilized with mineral fertilizers. On the other hand, Sapek and Kalińska [49] showed that after fertilization with liquid manure the amount of nitrogen released from the soil as a result of mineralization of organic substance was lower than from the soil fertilized with the same dose of mineral nitrogen. Similarly, Wesołowski and Durkowski [69] report that concentration of N, P, K, and Na in underground water was lower in objects fertilized with manure than in objects fertilized with mineral fertilizers. The highest loads of the analyzed nutrients that were leached with leachate water in the second period of the study should be linked to three facts: a lower dynamics of the uptake of these nutrients by more slowly growing vegetation, their higher supply in the soil on account of intensified biological activity, and the highest water leaching index. This last relationship is confirmed by many researchers. They claim that there is a positive relationship between the quantity of leachate water and the quantity of minerals leached from the soil [49,59,71]. The significantly higher quantities of nutrients leached with leachate water from arable land compared to the nutrients leached from the meadow community should be linked to the higher water leaching index and, according to Aarts et al. [72], lower content of organic substance content in the soil of arable land. The lower index of water leaching from meadow communities was a result of a substantial condensation of ground cover, as well as a significant accumulation of water by organic substance. That higher organic substance content in the soil in the meadow objects was also a significant factor in the retention of minerals, in other words in protecting them against leaching.

## 5. Conclusions

1.  The highest annual load of nitrogen was leached from arable land. It was two- to three-fold higher than the loads leached from the other objects.
2.  Total annual load of leached phosphorus was generally highest in the first year, and in arable land in the third year. On average, annual loads of phosphorus leached from arable land were twice higher than the loads leached from meadow objects.
3.  The highest annual leaching of potassium was recorded in arable land, and the lowest in the control. The difference between these values was over two-fold.
4.  The highest annual load of calcium was leached from arable land. Its magnitude was 1.5 times higher than that of the load leached from the minerally fertilized object and over two times higher compared to the other objects.
5.  The lowest annual load of magnesium was leached from the object fertilized with liquid manure, and the highest from arable land. In the latter case, this load was over two-fold higher than the loads from the other meadow objects.
6.  The annual load of leached sodium in meadow objects was similar and ranged from 3.2 to 3.8 kg·ha−1, whereas in arable land it was twice as high.

Basing on the performed study, one can arrive at the following generalizations:

1.  Concentration of macroelements in water depended on plant growth and microbiological activity in the soil. That concentration was positively correlated with the intensive plant growth and dynamics of organic substance decomposition in the soil.
2.  The load of minerals leached with leachate water from the soil was highest in the second period of the study. The magnitude of this load was affected by such factors as the amount of water moving through the soil and the intensity of organic substance decomposition in the soil.
3.  Grassy communities, compared to arable land, retained significantly more meteoric water and minerals, thereby protecting the soil against losing them.

On account of the quality of leachate waters in submontane and montane areas, it is recommended to reduce plow tillage in these areas. It is also recommended to use these areas as meadows and pastures, with moderate fertilization and rational use, i.e., two mowings or three grazings during the growing season.

**Author Contributions:** Conceptualization, P.K. and J.S.; methodology, P.K.; software, J.S.; validation, J.S.; formal analysis, P.K. and J.S.; investigation, P.K.; data curation, J.S.; writing—original draft preparation, P.K. and J.S.; writing—review and editing, P.K. and J.S.; visualization, J.S.; supervision, P.K. and J.S. All authors have read and agreed to the published version of the manuscript.

**Funding:** This research received no external funding.

**Institutional Review Board Statement:** Not applicable.

**Informed Consent Statement:** Not applicable.

**Data Availability Statement:** Not applicable.

**Conflicts of Interest:** The authors declare no conflict of interest.

**Appendix A**

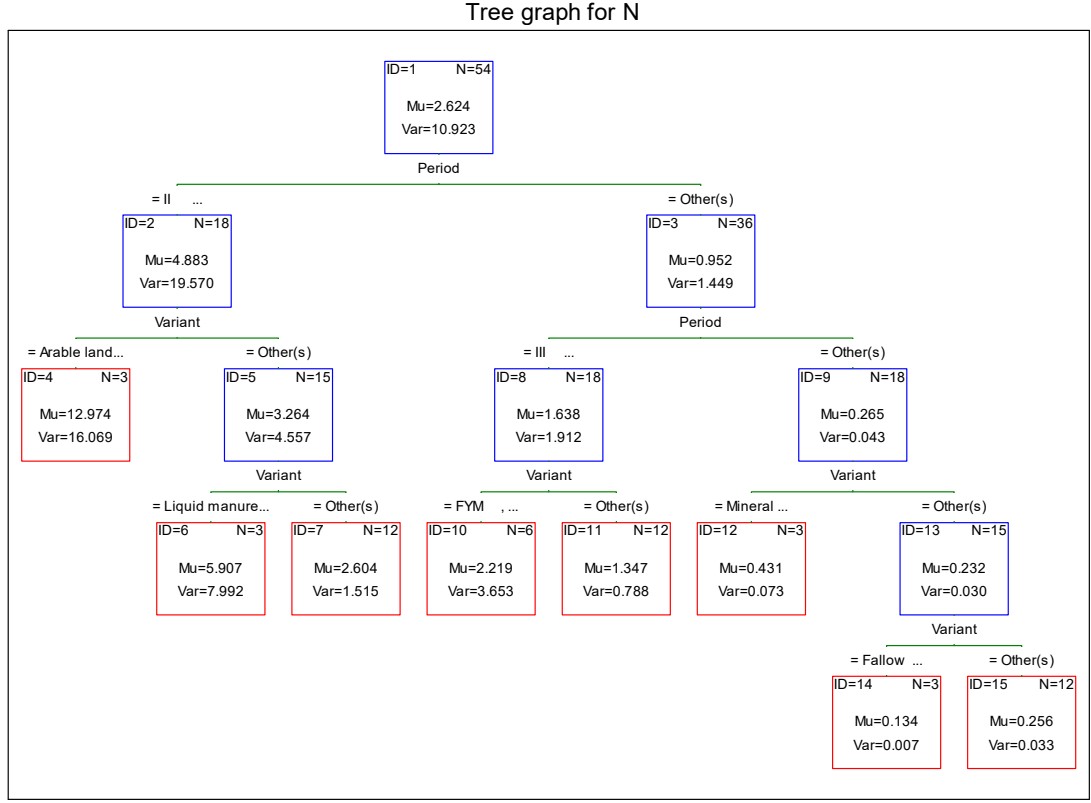

**Figure A1.** Factors determining nitrogen leaching to the ground water (Regression Decision Tree C&RT).

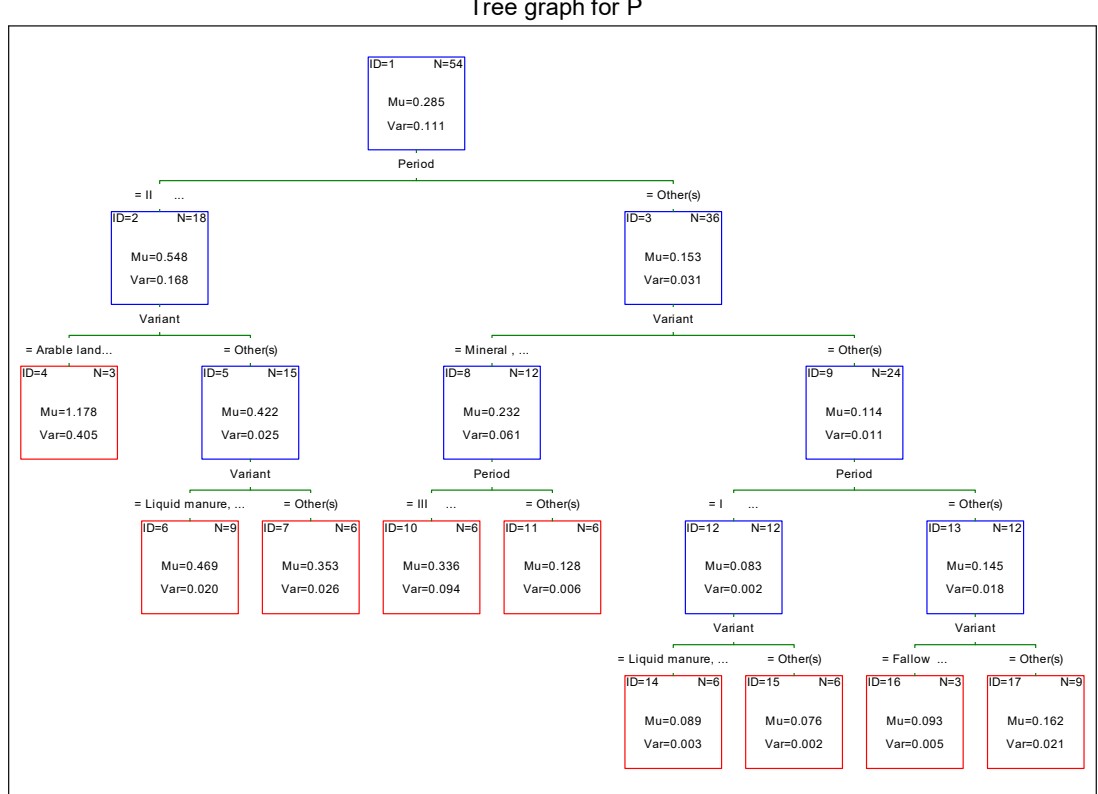

**Figure A2.** Factors determining phosphorus leaching to the ground water (Regression Decision Tree C&RT).

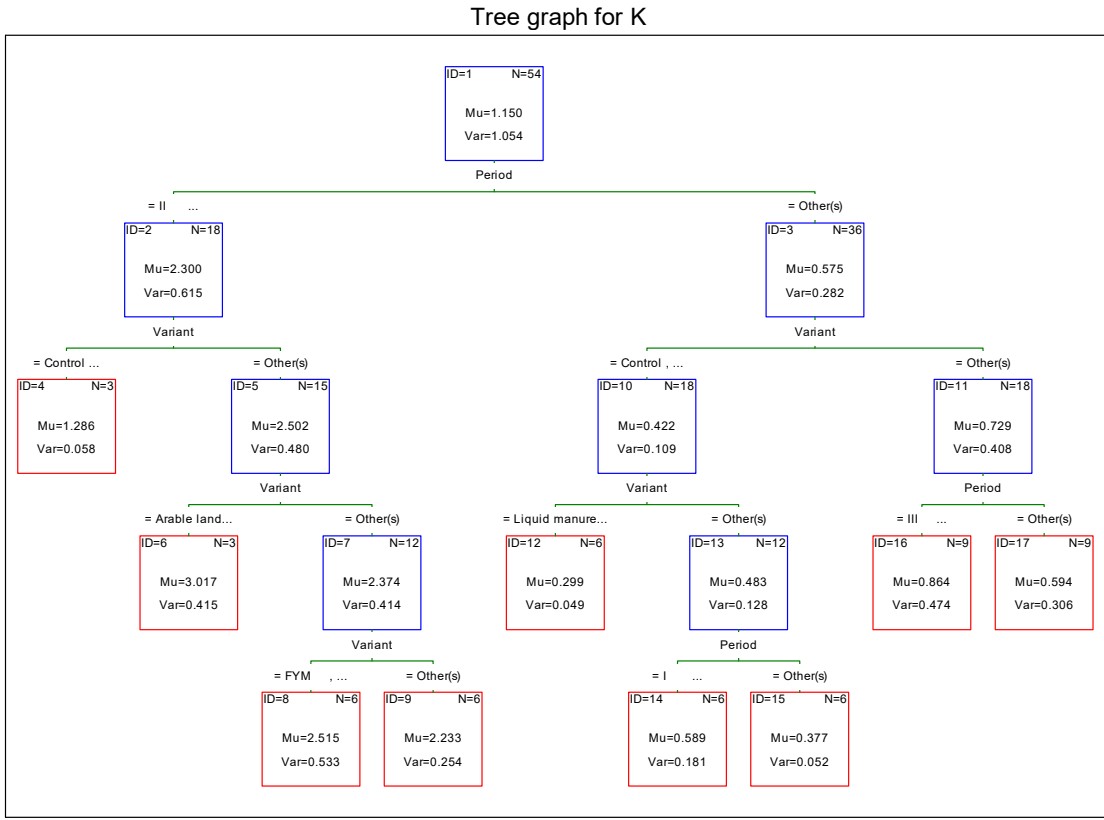

**Figure A3.** Factors determining potassium leaching to the ground water (Regression Decision Tree C&RT).

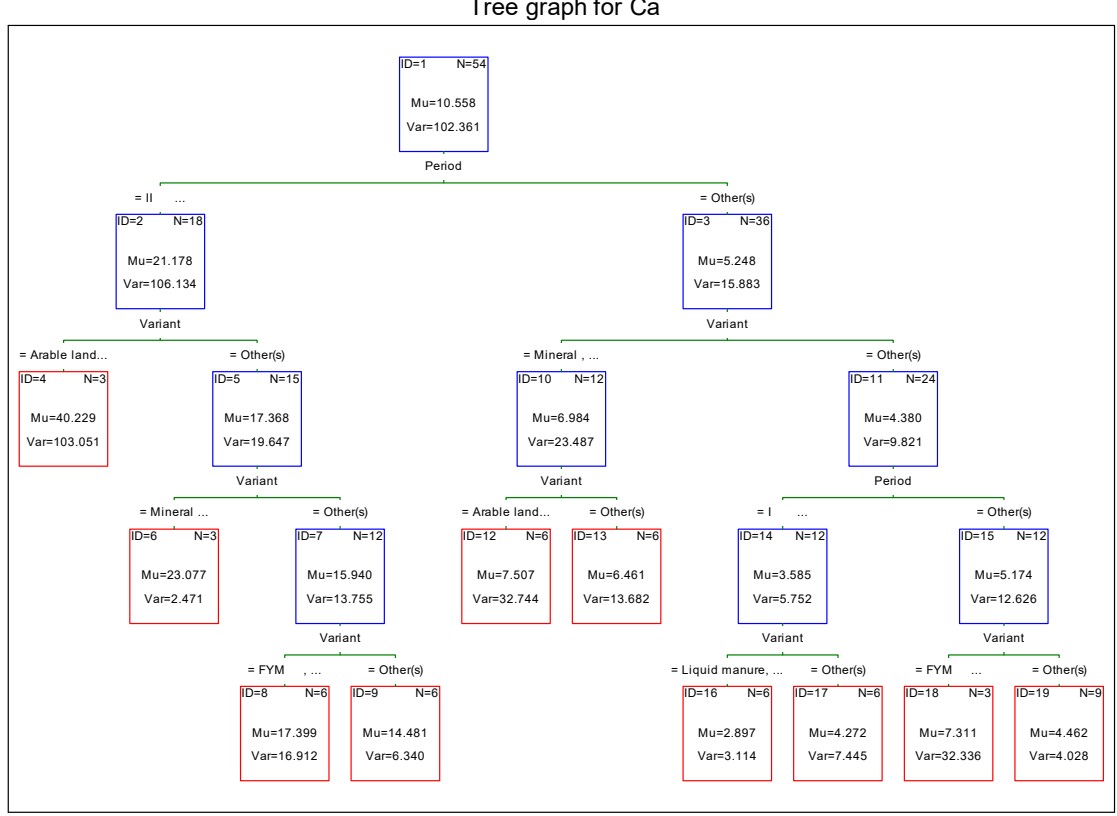

**Figure A4.** Factors determining calcium leaching to the ground water (Regression Decision Tree C&RT).

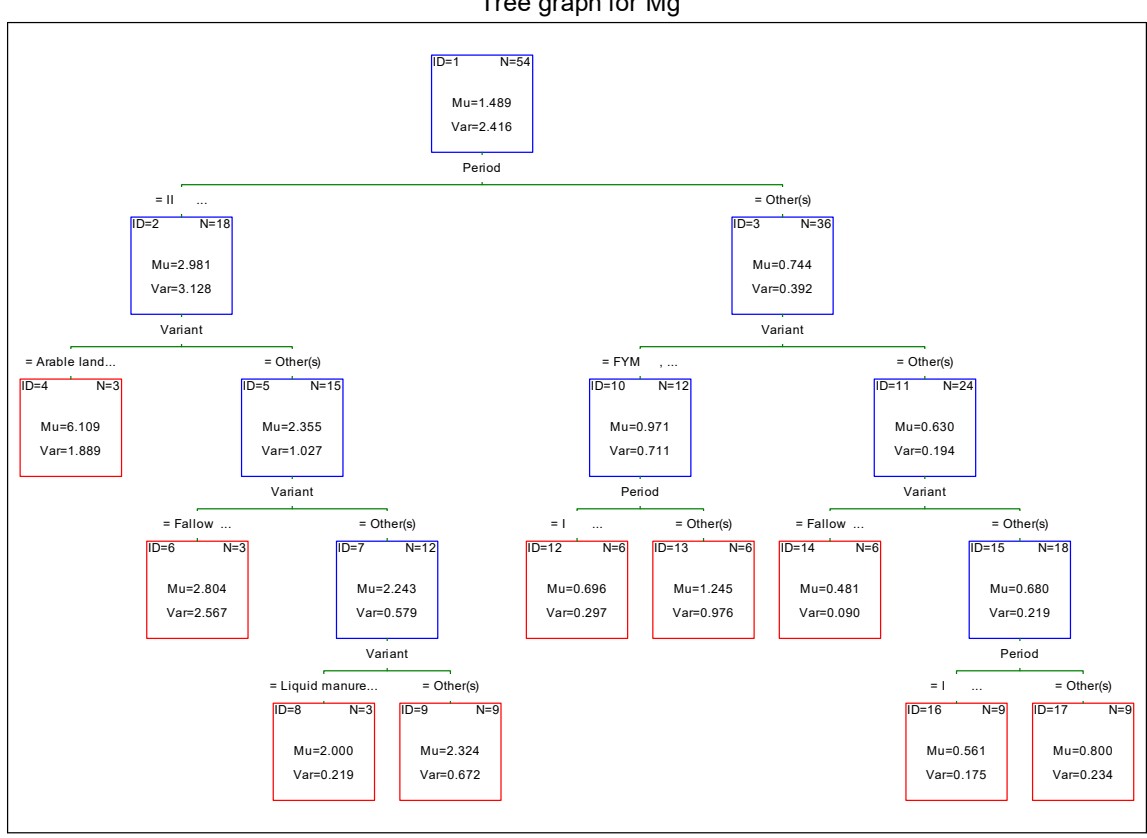

**Figure A5.** Factors determining magnesium leaching to the ground water (Regression Decision Tree C&RT).

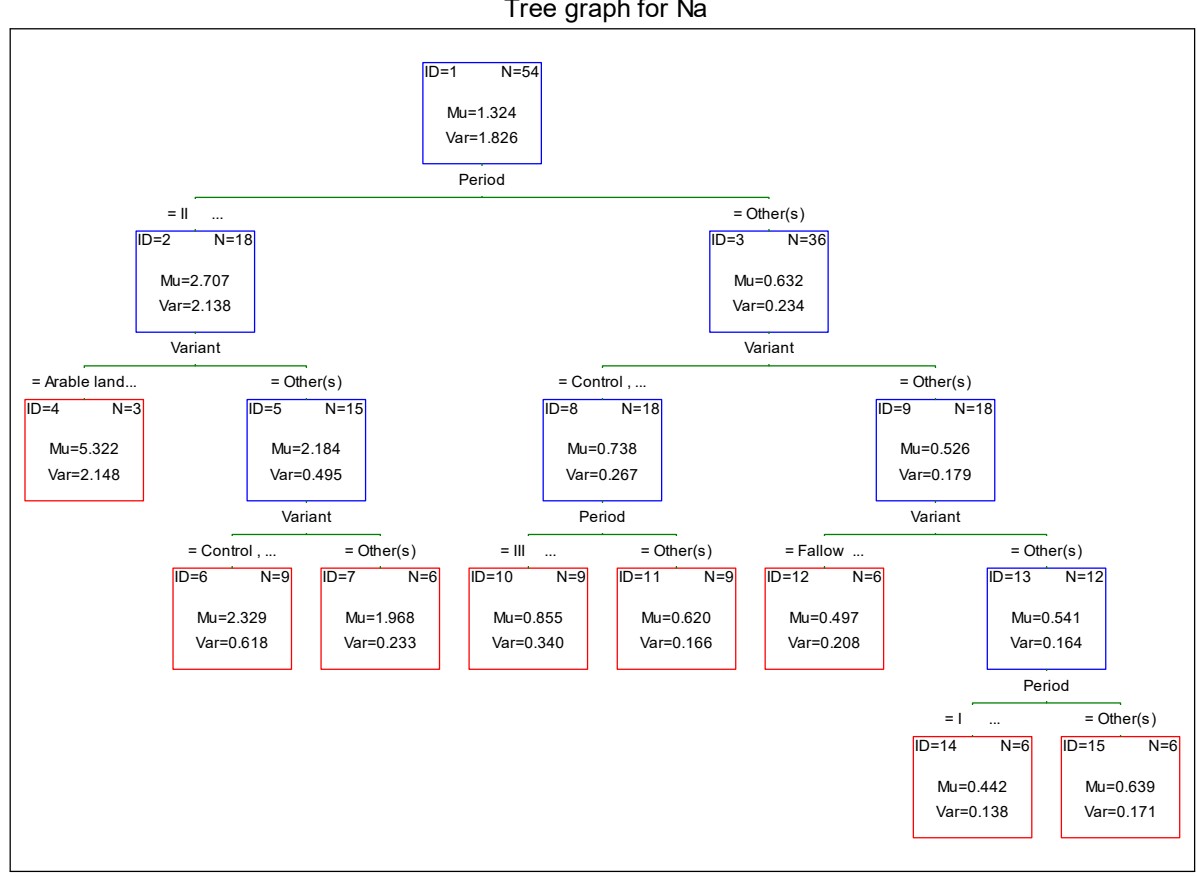

**Figure A6.** Factors determining sodium leaching to the ground water (Regression Decision Tree C&RT).

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
