# Peer review of "The Effect of the Manner in Which Montane and Submontane Areas Are Utilized on the Quality of Leachate Water"

_sustainability, doi:10.3390/su13116299_

Round 1

Reviewer 1 Report

Section 1 and section 2 will be joined because both establish the review of their research

Really, the review is very well, but authors should expose clearly the main goal of their research at the end of the new section one

In materials, they should explain why did they do these experiments and explain better their implications in the research

The methodology should be explain using a flowchart where the authors explain each phase and what is the results, they wants to get

I think the figures and tables should be in the text, and they must be discussed deeply.

What is the uncertainity of their experiments?

Conclusions should be rewritten

“one can arrive at the following generalizations”, really it is a conclusion of research study. I think generalization cannot be conclusions of one research

Author Response

Dear Reviewer,

Thank you for giving us the opportunity to submit a revised draft of the manuscript titled

“The effect of the manner in which montane and submontane areas are utilized on the quality of leachate water”.

We appreciate the time and effort that you have dedicated to providing your valuable feedback on the manuscript. We are grateful for your insightful comments on our paper. Authors highly appreciate all tips.

The changes and corrections have been made within the manuscript.

A point-by-point response to the review’s comments and concerns has been made in attached file.

Sincerely,

The authors

Reviewer 2 Report

The paper is well written, however there a few aspects that should be revised:

  1. I recommend the authors to present the tables in the text, not in the appendix.
  2. please present the limitations of the study, managerial implication, future research directions.
  3. please emphasize the link between the journal aim and the paper

Author Response

(The authors gave the same response as above.)

Round 2

Reviewer 1 Report

The authors answered all suggestions and improved the understanding of the research.

Author Response

Dear Reviewer,

Thank you once again for giving us the opportunity to submit a revised draft of the manuscript.

We appreciate the time and effort that you have dedicated to providing your feedback on the manuscript.

We are grateful for your insightful comments on our paper. Authors highly appreciate all tips.

We would like to take this opportunity to thank you for the effort and expertise, without which it would be impossible to maintain the high standards of the manuscript.

Sincerely,

The authors

Reviewer 2 Report

Dear author/s,

thank you for improving the manuscript. Now it can be considered for publishing.

Good luck!

Author Response

(The authors gave the same response as above.)
